# Emotional connectedness and adolescent help-seeking behavior for mental health problems: A situational analysis

**Katrin Häggström Westberg**[1], **Malin Eriksson**[2], **Mikael G. Ahlborg**[1]*

**1** School of Health and Welfare, Halmstad University, Halmstad, Sweden, **2** School of Social Sciences, Södertörn University, Södertörn, Sweden

* mikael.ahlborg@hh.se

## Abstract

Adolescent mental health problems are increasing globally, emerging during a developmental period marked by shifts in social relationships, identity formation, and growing autonomy in health behaviors. Although adolescents may access both formal and informal sources of support, many refrain from seeking support due to a range of individual and structural barriers. A deeper understanding of the social processes involved in help-seeking in adolescence is needed, particularly regarding the role of emotional connectedness and how it relates to the help-seeking process. The aim was to explore adolescents' perceptions of how emotional connectedness influences help-seeking for mental health problems. Twenty-four adolescents, aged 15, from two schools in the south-west part of Sweden, agreed to participate in focus group interviews. A Situational Analysis was carried out in multiple steps. The results show thatemotional connectedness, distinguished through emotionally close or distant relationships, either facilitated or hindered help-seeking intention based on how the relationships were perceived and what struggles were discussed. Emotional closeness represented something secure and predictable, but could be perceived as conditional and burdensome related to help-seeking. Emotional distance could be perceived as unconditional and alleviating, thus, facilitating help-seeking, but was also ascribed to relationships where help-seeking felt insecure and unpredictable. In conclusion, depending on the context, both emotional closeness and distance may facilitate or hinder help-seeking. The dual meaning of emotional connectedness in relation to help-seeking warrants further investigation.

## Introduction

The rising prevalence of reported mental health problems among adolescents constitutes a significant global public health concern [1–5]. Self-reported mental health problems have increased drastically among Swedish adolescents and exceed those

**Data availability statement:** The ethical approval received from the Swedish Ethical Review Agency allows the sharing of de-identified transcriptions of the focus group discussions in the Swedish language, upon reasonable request. This data sharing restriction protects the confidentiality of the participants. Requests may be sought by contacting dataskydd@hh.se at Halmstad University.

**Funding:** This work was supported by the Kamprad Family Foundation (Grant nr: 20223264 to MA). The funders had no role in study design, data collection and analysis, decision to publish, or preparation of the manuscript.

**Competing interests:** The authors have declared that no competing interests exist.

in other Nordic countries [6]. This trend is also reflected in a marked increase in mental healthcare contacts over the past decade, alongside a rise in prescriptions for depression and anxiety treatments [7]. Nevertheless, despite the escalating incidence of mental health difficulties, a considerable proportion of adolescents either refrain from seeking professional support or experience prolonged help-seeking processes [8–11].

Adolescence represents one of the most intense periods of life, marked by profound physical and psychological maturation alongside the social challenges associated with transitioning into adulthood [12]. This phase is commonly viewed as a critical time for the development of social identity, shaped both by increasing independence from established social relationships and the formation of new social ties. It is also a period where autonomous health behaviors emerge, which in turn can influence social behavior patterns and health trajectories well into adult life [13,14]. In this context, social networks play a significant role, as they can both buffer and intensify the immediate and long-term effects of mental health problems [15]. Given adolescents' heightened sensitivity to peer influence and social evaluation [16], these networks may shape the ways in which they handle mental health problems. Consequently, patterns of seeking or refraining from support may persist into adulthood and impact long-term well-being [17].

There is a need for more research on where, why, and how adolescents seek help, and the sources and nature of support available for young people in different contexts [17,18]. The sources of support for this group with mental health problems are not limited to health services but comprise both formal and informal sources [19]. In a review of barriers and facilitators in relation to help-seeking behaviors for common mental disorders, Aguirre Velasco et al. [20] identified a range of help-seeking options for adolescents, both formal (medical doctors, psychologists, psychiatrists, teachers, social workers) and informal (friends, family, sports coaches, and online communities). Much research has mainly focused on help-seeking in relation to formal sources of help, i.e., healthcare professionals, while less is known about under what conditions adolescents turn to informal sources, such as family and friends, for support [19,20]. There are studies indicating that young people with mental health problems commonly turn to family members and friends for support rather than professionals [19,21,22].

Studies have highlighted several barriers for adolescents' help-seeking, such as low mental health literacy with limited knowledge of mental illness and health services, and uncertainty about what constitutes a need for help, but also organizational barriers and siloed care [20,21,23]. Furthermore, previous studies have established how adolescents' help-seeking is hindered by negative perceptions of support services and the help-seeking process, perceived stigma, and, not least, confidentiality concerns and lack of trust in the help provider [11,17,20,21]. Conversely, family and school connectedness, positive perceptions about contact with professionals, and positive encouragement from support networks have been found to facilitate adolescents' help-seeking [11,20,21].

The notion of connectedness (including social, interpersonal, and emotional forms) and its significance for adolescent and youth mental health is well established in

recent literature on suicide risk [24], depression and anxiety [25], and wellbeing [26,27]. Emotional connectedness has been conceptualized as a qualitative dimension of social connectedness, referring to the perceived closeness, trust, and mutual emotional responsiveness within relationships. Whereas social connectedness broadly captures structural and functional aspects of relationships, including social networks, perceived support, and absence of social isolation [28], emotional connectedness specifically refers to the quality of these relationships, particularly feelings of intimacy, safety, and being emotionally understood [29]. The importance of nurturing social connections and battling loneliness is a global priority, and the potential societal health benefits are well documented [18]. Moreover, access to social support can help promote adolescent mental health [30] and may also function as a pathway to formal help-seeking, with informal sources acting as bridges to more formalized sources of support [19].

Research on help-seeking is often influenced by behavioral theories anchored in the belief that behavioral intentions are predicated by intention, as formulated in the Theory of Reasoned Action by Ajzen & Fishbein [31]. More recently, the Integrated behavioral model of mental health help-seeking (IBM-HS) was introduced by Hammer et al. [32], proposing a model that identifies the causal sequence of constructs influencing mental health help-seeking behavior whilst incorporating both determinants, beliefs, and mechanisms that translate into intention and ultimately help-seeking behavior. However, intentions seem to be reliable, but limited predictors of behavior, and even strong intentions translate into behavior inconsistently [33]. Recognizing that help-seeking encompasses many facets, the Network-Episode-Model [34] outlines help-seeking as a complex and not logical process, spanning layers of both social, psychological, and contextual factors. Any definition must encompass the whole spectrum within the help-seeking process. Therefore, for this study, the definition by Rickwood & Thomas is used, stating that "help-seeking is an adaptive coping process that is the attempt to obtain external assistance to deal with a mental health concern" [35] (p. 180). This definition incorporates both the temporal process, the type of support sought, the underlying concern, as well as the source, ranging from formal to informal and even self-help sources of help. Understanding the dynamics and challenges involved when this group of individuals seeks help from both formal and informal sources is imperative, not least, regarding the significance of emotional connectedness in adolescent help-seeking.

### Aim

The aim was to explore adolescents' perceptions of how emotional connectedness influences help-seeking for mental health problems.

## Materials and methods

### Overall study design

This study is part of a larger project that aims to investigate social capital in relation to help-seeking and adolescent mental health, undertaken in close collaboration with adolescents and key interest-holders in the southern part of Sweden [36]. In the project, the starting point was to explore the concepts of social capital and help-seeking in the context of mental health problems from an adolescent perspective. The present study incorporates, builds on, and extends from a first round of focus group discussions (FGDs) that were held. We employed a grounded theory situational analysis design [37]. Situational analysis (SA) is a development of grounded theory (GT) that recognizes the need to analyze the full situation of inquiry and not only the basic social process of the same (as in classical GT). Further, SA recognizes that sensitizing concepts are used as tools or guides when theorizing data. A sensitizing concept is a general, orienting idea or a preliminary starting point for thinking about empirical data, rather than a fixed, definitive concept.

### Study setting

Sweden has a population of about 10.5 million people (scb.se) and is considered a high-income country with a relatively low Gini Index, although it is rising [38]. The primary and lower secondary school in Sweden comprises both public and private schools, catering to pupils from grades 0–9, starting at age 6, followed by high school, which stretches over three

years. School satisfaction in Sweden has declined with reports of growing schoolwork pressure and reduced classmate support [39]. There is no governmental oversight of school absenteeism in Sweden, but based on compiled municipal data, the prevalence of school absenteeism (>20% absence) has doubled since before the pandemic in 2020 to about 10% [40]. Nationwide studies show that 15-year-olds report good communication with parents and a favorable family culture [39]. Qualitative studies report that adolescents describe access to a "safe space" with trusting relationships and predictable social networks as crucial for their well-being [41]. Participation in organized sports is common among early adolescents but drops significantly between 13 and 15 years of age, with approximately a third of 15-year-olds spending most of their free time sedentary. The low levels of physical activity coincide with high reports of multiple health complaints as well as high digital interaction [39], and research shows that Swedish sports coaches play an integral part in creating a social atmosphere that motivates continuation [42]. The current study was conducted in southwestern Sweden, in a region with a higher-than-average median income, lower unemployment rates, and a smaller proportion of residents holding higher education degrees [43].

## Ethics statement

All participants were given oral and written information about the study, adhering to the requirements for written informed consent, in accordance with the Declaration of Helsinki [44]. The researchers provided written information to the 9th-grade classes and allowed time for any queries or concerns raised by the pupils. The pupils were asked to return signed consent forms to their teachers within 10 days, allowing time for questions to be addressed by parents, teachers, and the research team. All participants were over 15 years old, which allowed them to consent to participation on their own. Participants were compensated with a cinema gift card ($15 value) upon completion of the data gathering. Ethical approval was received from the Swedish Ethical Review Agency [no. 2023-01531-01].

## Sampling of participants

A purposive sampling technique was applied to include schools that represented both rural and urban communities. Two schools were included, one private school in an urban setting and one public school in a rural setting. Both schools were of a similar size, with approximately 500–600 pupils. Recruitment was done from grade nine in both schools. Students aged at least 15 were eligible for participation. All students of age were eligible for participation, regardless of previous mental health history, after providing informed consent. Firstly, the principals of the two schools were contacted. The first and last authors from the research group visited the schools during school hours and informed the pupils about the study. Thirteen boys and 11 girls agreed to participate in the study, for a total 24 participants. This study focused on perceptions relating to emotional connectedness, help-seeking, and mental health problems. The inclusion criteria did not limit the sample to adolescents with documented experience of formal help-seeking, self-reported mental health problems, or a diagnosis. It became evident, however, that some participants had experienced mental health problems and both informal and formal help-seeking. The sample comprised public and private schools from socioeconomically diverse rural and urban municipalities and included an almost equal distribution of boys and girls.

## Data collection procedure

The data collection was conducted in two steps. Focus group discussions were chosen as the interview method to provide rich insights into participants' ideas, attitudes, and collective norms, while also revealing differing perspectives and interaction between a group of individuals [45]. In the first round, four FGDs took place in December 2023 at the premises of the respective schools. The FGD participants ranged from 5 to 11 participants; two groups were mixed gender, one group was boys only, and one was girls only. At both schools, all FGDs were held during school hours, in a secluded room. The interviewers did not know whether any of the participants had experienced mental health problems.

A semi-structured interview guide was used during the first round of FGDs. The initial aim was to have adolescents describe experiences and perceptions of social capital (i.e., resources in social relationships and networks) in relation to help-seeking and mental health. Questions relating to the exploration of social capital were asked, such as "what people or networks do you consider to be important for the well-being of adolescents in the 9th grade in general?" "What is it that makes them important?" and "If you are not feeling well mentally, what is it about people that makes you feel you can confide in them?". Creative cards with examples of general networks, network members, and social contexts (e.g., school, neighbourhood) were used as a tool to facilitate discussions. Questions on help-seeking were asked, such as "how do you seek help in regard to mental health", "where would you seek help", and "concerning the time aspect, when is the appropriate time to seek help and what time perspective is useful for investigating help-seeking among adolescents?" The adolescents were also asked to reflect on seeking help in digital versus physical environments. Two members of the research team led the FGDs, which were audio-recorded digitally. FGDs were transcribed verbatim. As it was difficult to distinguish individual voices within the fourth and largest FGD, the audio data was complemented with field notes and memos. FGDs lasted between 49 and 61 minutes. A second round of FGDs based on initial analyses of the first round was undertaken in May 2024. In these, the same participants were invited, but all groups were divided by gender into girls and boys. Four FGDs with a total of 19 participants (8 boys and 11 girls) were held, lasting between 39 and 64 minutes.

## Data analysis

Situational analysis [37] was used for an inclusive analysis of the various data collected during the two rounds of FGDs (Fig 1). First, an open coding of the FGD transcripts was done separately by the members of the research team (KHW, MA, ME). The codes and initial thoughts were discussed and compared. Second, social world maps were drafted by the research group based on the FGDs to chart the basic elements of inquiry, social capital, and help-seeking. Social contexts and arenas, both physical and social, and resources embedded in different relationships were mapped. During an analytical workshop, these drafted maps were then discussed, and additional, more in-depth situational maps were created to illuminate tension within relevant concepts, such as help-seeking and mental health. This step in the analytical process involved going back and forth in the material, in line with basic principles in GT, which brought attention to particular sensitizing concepts [37]. Sensitizing concepts of interest were emotional connectedness, social capital, and help-seeking, and emotional connectedness emerged as the main concept of interest. Third, questions and hypotheses that arose during the second step of analysis regarding emotional connectedness and help-seeking were compiled into a short questionnaire. As an introduction to the second round of FGDs, the adolescents were asked to respond to the statements in the questionnaire. Positional maps had also been prepared for the respondents where they had to mark various people relating to the themes of emotional connectedness and help-seeking. The positional maps were used to enable discussion around the sensitizing concept of emotional connectedness. Fourth, subsequent to transcription of FGDs, KHW, MA, and ME conducted open coding of the second round of data, involving FGD transcripts, questionnaire responses, and situational maps. Fifth, the authors engaged in focused coding of first and second round data with emotional connectedness as the concept of interest. This step included coding sub-groups independently to allow for exploration of gender and school differences. In the sixth step, patterns and themes were discussed at recurrent analytical workshops, which resulted in the construction of categories. Moreover, a model of the main categories and concepts was constructed, relevant to how emotional connectedness relates to help-seeking for mental health problems among adolescents, with sub-categories covering the circumstances surrounding help-seeking and emotional responses. To strengthen trustworthiness, quotations are used in this paper to illustrate the findings. The quotations were translated from Swedish to English by a native English-speaking professional translator and were edited only slightly to improve readability, without losing the essence and meaning of the quote. When presenting quotes in the results section, I1 and I2 refer to the moderators, while the other numbers refer to different participants.

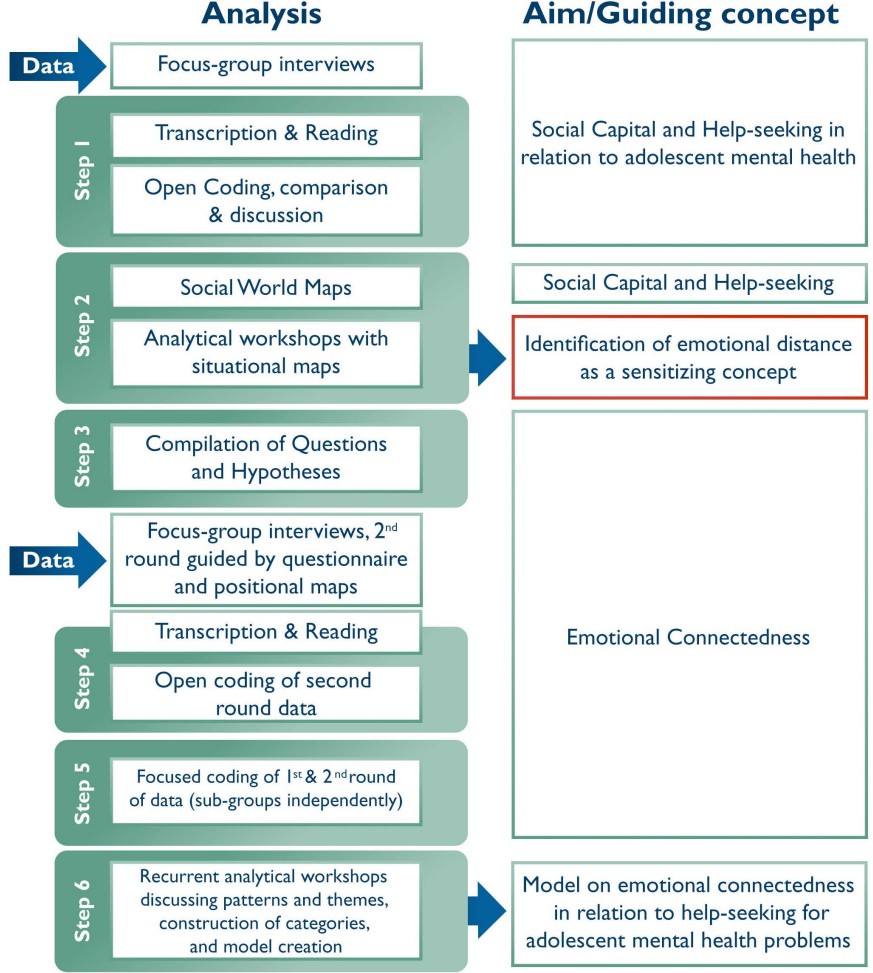

**Fig 1. Flow scheme of Situational Analysis.**

## Results

The results are based on how emotional connectedness is perceived in relation to help-seeking shared by the adolescents. At the time of the FGDs, they were approaching a significant transitional period in their lives. Being just months away from leaving the familiar social environment of secondary school and most entering high school after the summer, they were aware of the social changes that this would bring. Simultaneously, they were at various stages of the emotional separation from their parents that naturally occurs in early- to mid-adolescence.

When discussing help-seeking for mental health problems, the adolescents associated help-seeking with a range of social situations. Depending on the nature of the situation and the perceived severity of the problem, both formal and informal sources were considered. Emotional connectedness ranged from close to distant and was linked to different types of relationships, thereby influencing help-seeking intentions. Being emotionally close was primarily associated with informal relationships, such as family and close friends, whereas emotional distance was more often described in relation to formal relationships, such as counsellors or mental health professionals.

Both closeness and distance were described in divergent ways in help-seeking contexts. Close relationships were perceived as either emotionally secure and predictable, facilitating help-seeking, or as burdensome and conditional, hindering

it. In contrast, distant relationships were experienced as insecure and unpredictable, reducing the likelihood of help-seeking, or as unconditional and relieving, thereby enabling it. The model (Fig 2) illustrates how adolescents described relationships in relation to help-seeking from the perspective of emotional connectedness. Perceptions of relationships prompted emotional responses that either facilitated or hindered help-seeking, depending not simply on closeness or distance, but on how these were experienced in the context of seeking support.

## Emotional Closeness

The adolescents described emotional closeness with reference to mainly parents, siblings, close friends, and relatives. Emotional closeness in general meant feeling close to someone you trusted and that you felt connected to.

## Secure and predictable

When the adolescents in the FGDs reasoned about what facilitates help-seeking in emotionally close relationships, they conveyed feelings of being understood, looked after, and at ease. Emotional closeness implied a sense of security and predictability in challenging situations. They described that the immediate family knows you and your backstory through shared experiences, and this was a key factor in the participants' willingness to open up and feel at ease. Confiding in parents, siblings, and sometimes grandparents, mostly felt safe. When help-seeking for severe mental health problems was discussed in the FGDs, the participants highlighted that parents were primarily seen as a trusted choice. The adolescents expressed that help-seeking was facilitated through personal knowledge, and that for them to confide and to seek help, the other person had to be familiar with their background and situation. Most agreed that you would rather talk to somebody who knew you than someone who didn't.

Older siblings were also described as relatable figures who made you feel understood, as they had recently been through similar experiences.

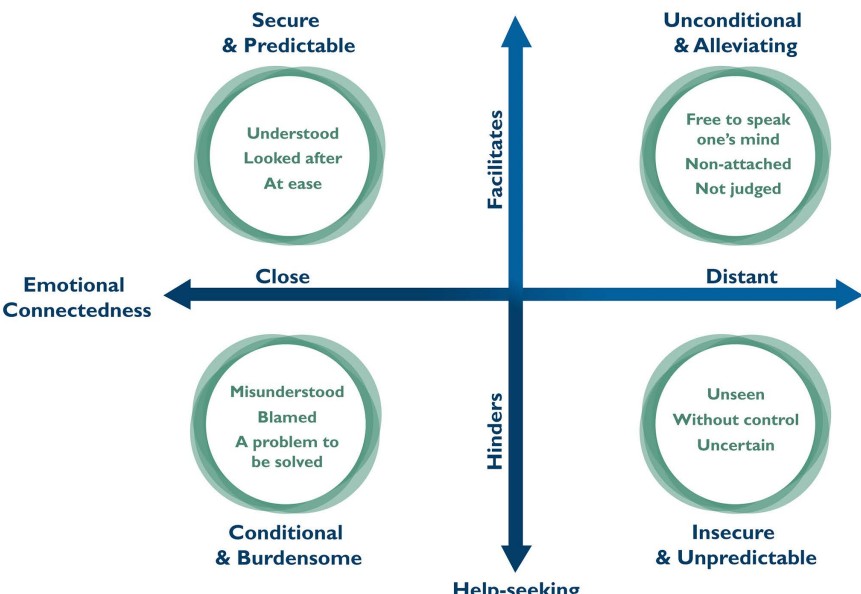

**Fig 2 How emotional connectedness was perceived in relation to help-seeking for mental health problems.**

*7:"Well, I mainly chose my sister because I know that she is the kind of person I can go and talk to if I feel bad… mmm and I also know that she can, like, come to me and tell me… so me and her are very close… eh… so I kind of know that we tell each other everything... I have no secrets from her... she is one of the most important people in my life..."* (FGD 1st round, girls)

When discussing trust, parental oversight, and help-seeking, most described that they felt comfortable sharing their smartphone location with their parents. Rather than feeling intrusive or controlling, this was perceived as reassuring – in case "something bad" happened, as a sign of affection and positive parental involvement. It was described as being looked after by their parents. This kind of parent-child trust, however, came with an agreement that parents would not call to check in, as long as the adolescents were where they said they would be. They perceived their parents to be not only trustworthy but also problem-solvers, which was considered a strength because solutions or advice were available from those who possessed "wisdom" and life experience. A gender difference appeared in their accounts of what parents were thought to be able to assist with. Boys primarily mentioned receiving help through parents' human and economic capital, whereas girls expressed getting emotional support. Both boys and girls, however, stressed that emotional closeness characterized the parental bond, which gave them a sense of security and predictability in parents' efforts to make them feel looked after.

Friends presented a predictable source of support by offering secure emotional relief, someone to talk to, and distractions, when needed, by suggesting enjoyable activities. Some adolescents considered peers relatable and capable of empathizing with their struggles to a greater degree than parents or siblings. Linked to emotional closeness, trust, which was emphasized as a precondition for help-seeking in general, was most strongly articulated with reference to friends, as trust was the foundation for friendship itself. Boys described a certain forum for exchange of thoughts that was not mentioned by girls – online gaming. While gaming online, they would talk, often in small groups, to fellow gamers and friends about issues other than the game. They particularly described how "real talks" would transpire late at night.

*I2: What do you all think about the whole idea of hanging out online? 4: Yes 3: It's cozy... 4: It is fine... you can talk a lot there... 6: You actually can... 4: You can, like, say what you want... 6: It's open conversations actually...* (FGD 2nd round, boys)

## Conditional and burdensome

The main reasons why emotional closeness was perceived to hinder help-seeking were adolescents' concern for loved ones, and the sense of being misunderstood, blamed or a problem to be solved. The participants expressed how disclosing personal issues, such as mental health problems, would place a burden on the receiver. Some feared that parents would blame themselves and that by keeping information from them, they could protect their parents from emotional strain. They would also avoid confiding in parents if they anticipated a strong emotional reaction from the parent that could include distress, sadness, anger, or blame.

*6: and then there's a lot like... things that you don't want to, or don't dare to, say to your parents because you are afraid that they will get angry, you're afraid they will be, like, yeah, but "if I say this then I will have to be home by nine o'clock every day, like" because that has happened to me and it's not fun...* (FGD 1st round, girls)

Because they did not know how the parents would react, they were apprehensive about reaching out. This was also mentioned in relation to sports coaches or other adults with whom they felt emotionally close. A similar perception existed in friendships, where they were cautious of creating a sense of obligation in their peers, knowing that close friends might

understand but not necessarily be able to help in a meaningful way. Thus, there was a tendency to cope with difficult issues on one's own.

*6: Yeah... it's like this... you shouldn't share so much if you're feeling down and spread it to others... I2: Why is that? 6: Then they are pressured, they might feel that they have to do something... I1: I see... I2: So, it's like you don't want to burden anyone with... 6: Exactly... I1: Okay... not worry anyone, or? 2: Yes, exactly.* (FGD 1st round, boys)

Another fear that was described related to parents' willingness to solve the adolescents' problems. Some depicted situations where they only wanted someone to listen to them, and perhaps receive advice, but not for someone to solve their problems. Whenever parents showed these tendencies, for example by initiating contact with others, adolescents expressed a loss of control and a sense of blame being put on them for not solving their own problems.

*4: If you just want to, like, talk... then they [parents] want to do something about it... and that might not be so good.* (FGD 1st round, boys)

A general perception was that adults could not understand young people today, and this hindered help-seeking as well.

*3: No, but I think that parents, like, they can be sadder, in the case when it's serious and they can start to blame themselves. Like, it is harder to talk to your parents, I think. I1: Is it that you don't want to burden them? 3: Mm. It also feels like they cannot quite understand.* (FDG 2nd round, girls)

The adolescents expressed tension when a sense of emotional distance arose in an otherwise close relationship. Feeling misunderstood led to withholding information or refraining from help-seeking, especially if the struggles were of a sensitive nature. Tensions were also present in friendships, but they seemed to be more related to loyalty and confidentiality. An example was discussed where, if a friend, a peer, came to the young person for support, adults would not be involved, unless the friend was okay with it, even if quite severe mental health problems were the reason for seeking help.

## Emotional distance

Emotional distance was mainly described with reference to people the adolescents did not meet daily, such as healthcare professionals, the school nurse, a sports coach, or the parents of friends. Teachers and mentors in the school were also mentioned, along with friends made online. Emotional distance generally implied the absence of formed trust and not knowing much about the other person. Gender differences were apparent in this category; girls voiced the stereotypical impression that boys would not seek help nor discuss emotional matters among themselves or matters relating to mental health. While the boys expressed awareness about this stereotype and "macho culture", they emphasized the importance of boys sharing their emotions and struggles.

## Unconditional and alleviating

The adolescents voiced that emotional distance in a relationship could sometimes be interpreted in a positive light. Linked to an unconditional and alleviating emotionally distant relationship were feelings of being free to speak one's mind, being unattached, and not being judged. With no or little emotional attachment, they felt greater freedom and a sense of relief when they were able to share their thoughts with someone, knowing it would not impact their relationship. Thus, sometimes emotional distance was a good thing, providing an opportunity to voice one's problems more easily.

*2: Yes, I wrote to an online friend because we did not know each other at all, but we texted quite a lot, and it was actually very nice because... like, he didn't know who I was and I didn't know who he was. But we could still write like a lot*

*without knowing who we were, like. I1: What do you think made it easy to write despite not knowing... that you didn't know each other before? 2: Because he wrote in a very understanding way. Like, really nice and like... yeah but in a good way. 1: Maybe if they don't know anything about you, it might be easier to open up, but, like, they can't judge you. Because he doesn't know who I am and then I don't really care what the other [person] might think of me because he doesn't know me.* (FGD 1st round, girls)

The participants expressed that emotional distance in a relationship sometimes facilitated openness and help-seeking. This was valid for informal relationships and social contacts. It was also applicable to formal contact with healthcare professionals. In these cases, emotional distance was described as a good thing when somebody listened, and there were no or few expectations on the part of the listener. Contrary to the reports on help-seeking being facilitated by personal knowledge, some mentioned that anonymity facilitated help-seeking. Anonymity made it feasible to have somebody only listen and not pass judgment. Situations when this was mentioned were friends and contacts made online, and professional support centers for anonymous calls or chats.

*5: But I know some people who have... or, like, this is how you write with these, I don't know what it's called... 4: Like BRIS?[1] 1: Yes... *laughter* I2: that's right 4: And you are completely anonymous there... I2: Exactly. 4: That can be nice because then it doesn't go anywhere, it stays there.* (FGD 2nd round, girls)

1A Swedish Organization for Children's Right in Society

There were also comparisons of school personnel and more emotionally distant healthcare professionals, and how the latter would be easier to seek help from.

*6: Yeah, I think it is nice to talk, like, I mean, really talk to a person... and, like, with somebody you don't know... like a counselor, not like the school counselor because they will not... it feels like they will look down on you when you go through the corridors, so then I felt that it was nice to go to a counselor that I didn't know.* (FGD 1st round, girls)

Relationships with their friends' parents were also mentioned as potentially conducive to help-seeking, as they were not as emotionally close as one's own parents. More emotionally distant adults could serve as confidants, and there was a lower risk that friends' parents would judge or reprimand.

### Insecure and unpredictable

Emotional distance could also imply that the relationship was insecure and unpredictable. In these instances, emotional distance was described as a hindrance to sharing information and seeking help. Adolescents expressed that, in such relationships, they had feelings of lacking control, and they felt unseen and uncertain.

Trust was a recurring aspect of help-seeking, and it was voiced that a lack of trust increased feelings of uncertainty. When trust in general was discussed during the FGDs, they stated that few people that could be trusted, not fellow adolescents, nor adults in general Mentors and school staff were specifically mentioned as not being trustworthy, and this was mainly brought up by girls. Many expressed concern about confiding in school staff, stating that disclosures would not be kept confidential. Mentors were said to share information between them without considering confidentiality.

*2: I wouldn't trust the teachers not to share things– but perhaps some of my relatives [can be trusted]. I: Okay, why not the teachers? 2: Because when you tell them things they pass it on.* (FGD 2nd round, girls)

*I1: What is it that makes you not trust adults? 3: But it's like in school, you have experienced that a teacher has said like "you can tell me, I won't tell anyone". Then, another teacher knows and asks you how you are. 6: Yeah, they talk with each other an awful lot...* (FGD 1st round, girls)

As they thought that few adults could be trusted, the adolescents were left with a sense of uncertainty, and relationships were seen as insecure and unpredictable, which did not facilitate help-seeking. A lack of knowledge and understanding of issues relating to mental health was expressed. They admitted to having difficulty knowing when to seek help for mental health problems, i.e., assessing severity and length of experienced symptoms. Suicidal thoughts or behavior were said to warrant contact with adults, informal or formal, but, apart from this, the defining boundaries for mental health problems were unclear as was the time-point when to seek help.

> 1: It's weird to seek help if you're just feeling a bit low. It's a different thing if you are suicidal and consider self-harm and things like that. I1: What is it you're saying? Do you want to elaborate? 1: No, or yeah, like if you want to end your life then there is that limit. I1: and before that? 1: No, that I don't know. I haven't felt mentally unwell. Or I don't know, have I? I mean how do you know if you are mentally unwell? (FGD 2nd round, boys)

It seemed that the link between emotional distance and help-seeking was dependent on the severity of the issues that adolescents discussed. When problems were of a more severe character, formal sources were seen as a viable option, since emotionally distant relationships could be considered less burdensome and more comforting, carrying a sense of relief.

## Discussion

### Summary of main results

Emotional connectedness emerged as a key theme during the SA, offering insight into adolescents' reasoning regarding help-seeking in the context of mental health problems. Based on these findings, emotional connectedness can be conceptualized as a dynamic relational state that encompasses both affective experience and cognitive appraisal and is continuously shaped through ongoing interpersonal interactions. Related to help-seeking, the meaning and implications of emotional closeness or distance depend on adolescents' interpretations of these experiences, which in turn are influenced by contextual and social factors. Consequently, help-seeking is a complex and dynamic process, shaped by adolescents' perceptions of their social environment. This suggests that adolescents continuously re-evaluate the meaning of emotional connectedness when contemplating or reflecting on help-seeking in relation to mental health problems. Aligning with Rickwood & Thomas [35] definition of help-seeking as an adaptive coping process, the present findings suggest that its adaptive nature is contingent on adolescents' relational evaluations, with emotional distance and closeness differentially shaping not only whether help is sought, but from whom and under what conditions. One key finding is that emotional distance can facilitate help-seeking through more than just professional confidentiality. Despite the lack of intimacy, closeness, and built-up trust, generally associated with facilitated help-seeking [46], our results unveil a nuanced mechanism in which emotional distance may offer a safe outlet for disclosing emotional and social struggles without fear of consequences beyond those addressed by professional confidentiality. On the other end of the spectrum, emotional closeness often meant a secure and predictable bond. Still, these close relationships could also be experienced as conditional or burdensome, depending on the situation.

### The paradoxical role of emotional closeness for adolescents' help-seeking

The results clearly illustrate that emotional closeness may facilitate adolescents' willingness to seek help but can also be a barrier to help-seeking. Emotional closeness, often attributed to family members and friends, generally offers a safe environment where adolescents feel understood, looked after, and at ease, which facilitates help-seeking. The significant role of easily accessed and trusting, informal relationships in facilitating help-seeking for mental health issues has been underlined by others [47,48]. Our results are in line with quantitative findings [49], indicating that emotionally close relationships (e.g., family and friends) are perceived as the first option when problems are not too serious, while formal and emotionally distant relationships (such as with healthcare professionals) are perceived as a help-seeking option when problems

are more severe. In a review of the role of informal sources of help in young people's involvement in professional mental healthcare, Lynch et al. [47] found that informal helpers also can function as important intermediaries to formal sources of help, e.g., mental health services, and provide continuous practical and social support. The role of friends and family as mediators in help-seeking becomes even more critical given that many adolescents experiencing severe depressive symptoms or engaging in self-harm do not seek help at all [49,50]. From a theoretical perspective, our findings show that the help-seeking process is influenced by multiple contextual and individual factors, in line with the network-episode model, emphasizing help-seeking as a dynamic social process shaped by interactions between individuals and their surrounding networks [34].

Our results show that emotional closeness could also encompass a sense of not wanting to burden a significant other with one's problems. As such, closeness becomes a barrier to seeking help. This perception may further encourage self-reliance, i.e., a preference for solving problems on one's own rather than seeking help, which has been identified as a major psychological barrier for young people to seek support for mental health issues [21,23,48,51]. In emotionally close relationships, expectations for help and support are often high, as family and friends are assumed to care for and assist one another. However, when these expectations are not met, individuals may be more likely to feel misunderstood or even blamed. At the same time, strong expectations of support within families, particularly from parents toward children, can unintentionally lead adolescents to feel like they are a "problem to be solved". This perception may reduce their willingness to seek help. Additionally, young people may avoid reaching out because they do not want to burden others or are concerned about the impact their problems might have on those around them. Previous research has similarly identified these concerns as significant barriers to help-seeking for mental health issues among adolescents [21,52].

Interestingly, our results do not highlight stigma as a strong barrier to seeking help for mental health problems, something which has been found in other studies [21,47]. In another study by Hellström and Beckman [53], consistent with the current prevalence and awareness of mental health issues among young people in Sweden, adolescents reported that shifting societal norms and attitudes have reduced the stigma associated with seeking help for mental health problems. However, what is found instead is self-stigma, self-blame, and internalization of negative stereotypes [53]. Moreover, adolescents describe several sources of mental health problems, such as dealing with the challenges of meeting expectations, constant comparison with peers, and difficulties maintaining healthy social relationships. Hellström and Beckman [53] advocate for the training of practical problem-solving skills among adolescents to enhance their resilience and independence and emphasize the importance of ensuring the inclusion, availability, and visibility of influential adult role models, both emotionally close and distant. While it is crucial to develop emotional regulation and problem-solving competencies, a significant challenge arises when a problem appears to be minor or to have been resolved, yet the adolescent continues to experience substantial emotional distress. This residual emotional strain can persist for a period in which adolescents may be particularly in need of support. Our findings in the current study indicate that adolescents also consider the contextual circumstances or the severity of their mental health challenges when exploring options for help-seeking. While adolescents may turn first to a close friend or a parent for some concerns, they may find it inconceivable to do so for other problems. This highlights the importance of a rich social network with multiple adult role models where adolescents can seek guidance.

We found gender differences in the reasoning around help-seeking in emotionally close relationships. Previous research shows that boys consider help-seeking to be a sign of weakness [21]. Our findings partly support this, as the boys described a slight hesitance to explicitly seek help and an attitude of "it will get better as time goes by". However, they emphasized the need for boys to communicate their feelings in general and seemed willing to do so themselves. The girls, on the other hand, perceived boys as incapable of communicating emotions and not receptive to girls' help-seeking. Related to this, the review by Kågesten et al. [54] highlights that young adolescents across diverse cultural settings often internalize norms that uphold gender inequalities, with parents and peers playing a crucial role in shaping these attitudes. Such findings emphasize the importance of acknowledging social and interpersonal influences when promoting

help-seeking competency. Our findings reveal gendered stereotypes reinforced by societal norms. Changing such deeply ingrained norms requires a collective effort, involving families, communities, and institutions, to foster an environment where emotional expression is normalized and encouraged for all genders [54]. Early, context-specific interventions that target social environments and challenge normative beliefs are essential to building help-seeking competency in adolescent boys and girls.

### The paradoxical role of emotional distance for adolescents' help-seeking

Similar to emotional closeness, emotional distance can either facilitate or hinder help-seeking, depending on how adolescents interpret its meaning in a situation. Our results illustrate that seeking help from more emotionally distant sources, such as healthcare professionals, school counselors, friends' parents, and sometimes teachers, can be perceived as alleviating and unconditional, as long as confidentiality is ensured. Thus, relationships characterized by emotional distance may enable the adolescents to freely speak their minds without feeling judged or like a burden, while maintaining control over the information that they share. Similarly, the study by Radez et al. [21] shows that adolescents who perceive professionals as non-judgmental, respectful, and good listeners are more likely to seek help from formal sources. Our findings concur with this and add that the lack of emotional attachment in relationships can offer adolescents a sense of freedom by not having to worry about burdening a significant other. Moreover, our findings suggest that the opportunity to be anonymous and talk to professionals who adhere to confidentiality helps create a safe environment for sharing mental health issues. According to participants in our study, informal online friends can function in the same comforting way, the difference being that advice or support is not necessarily expected. This finding gives nuance to general assumptions that mutual trust and familiarity, or professional confidentiality, are prerequisites for help-seeking [46]. Instead, it suggests that, under certain conditions, emotional distance may foster a sense of situational safety that is distinct from, but complementary to, professional notions of confidentiality. The dual role of emotional distance revealed here, highlights the importance of contextual factors, including the nature of distress, prior help-seeking experiences, and the perceived availability of supportive others. Given an arising tendency for self-disclosure in emotionally distant relationships, and the increasing integration of digital communication into adolescents' everyday lives, these findings point to a need for further research into how online and peripheral relationships may serve as informal support systems. In particular, exploring how adolescents navigate and assign meaning to emotional distance in different contexts could provide valuable insights for the development of more accessible and responsive mental health support strategies. The advantages of anonymity in adolescent help-seeking have been investigated in the online setting [55], an arena that offers a multitude of help-seeking opportunities, both informal and formal. In general, the online arena is highly accessible, with low thresholds for contact, and adolescents can maintain a higher level of control in their help-seeking by way of anonymity [55]. Still, it is important to note that cyberbullying and grooming occur in the highly unsupervised online arena and can be potentially dangerous for vulnerable adolescents [56]. Therefore, it is important to educate adolescents also in digital literacy and increase awareness about what constitutes safe online help-seeking options.

Our results also illustrate how relationships characterized by emotional distance can evoke a feeling of uncertainty and not being seen, reducing willingness to seek help for mental health issues. In line with this, previous research has identified that experiences of their problems not being taken seriously by professionals become a major barrier for future help-seeking among young people [46]. Our study adds that breaches of confidentiality constitute a significant barrier to future help-seeking behaviors, not only for the affected individual but also for their broader social network, as such breaches erode generalized trust in the person or profession responsible for the violation. For instance, adolescents expressed how negative experiences led to concerns about the trustworthiness of teachers and school mentors. Aron Vallance [57] has elaborated on the ethical, legal, and clinical dilemmas surrounding the disclosure of confidential information, where a key consideration in maintaining confidentiality is preserving the trust of adolescents who experience mental health problems.

Previous research indicates that adolescents perceive a decline in social support from teachers when they transition to secondary school, potentially due to the involvement of multiple teachers, which complicates the development of personal relationships [58]. Our findings show that the support culture can differ between schools, and sensitive information from pupils may be passed on between teachers, if the culture allows. A study by Forsberg et al. [59] finds that one prerequisite for seeking support from teachers is a well-established relationship built on trust. Another key aspect is that teachers are presented as one source of support for pupils that complements other, more formal, sources of support.

## Implications for practice

These results increase our understanding of the challenges involved in adolescents' help-seeking for mental health problems and have important implications for both formal and informal help providers.

Confidentiality emerges as a crucial prerequisite for young people to seek help and needs to be acknowledged, especially among school personnel and healthcare professionals. When adolescents perceive that confidentiality is uncertain, help-seeking is hindered. Since lack of trust is a major barrier to seeking help, ensuring confidentiality should be a priority for adults meeting young people with mental health problems. This may, in some instances and settings, be complicated with regard to laws on the obligation of reporting. Our results indicate that information about pupils is shared between teachers and other staff members in schools, which damages trust and hinders help-seeking. A clear policy and discussion among colleagues regarding information sharing, roles, and responsibilities for the various professionals, such as teachers or school counselors, is paramount. If confidentiality cannot be secured for any reason, this must be clear and must not be promised. At the very least, teachers should inform pupils, and certainly ask for their consent, before sharing information with others. In addition, schools and other settings where adolescents spend time should provide clear information about available sources of confidential help. To complement existing formal professional services, our findings highlight the potential value of low-threshold, emotionally distant support options. These may include anonymous services, but more importantly, options that are highly accessible and relationally low-risk, thereby encouraging self-disclosure and facilitating help-seeking without fear of repercussions.

With regard to informal sources of help and support, our results clearly illustrate the indispensable value of parents and close family for adolescents. These family members most often represent the first help-seeking option. Depending on the response received from those closest to them and whether the adolescent feels understood, blamed, or misunderstood, significant others can be important bridges to more formal sources of help when needed. Further, the results expose that parents and close family might need to be able to hold back on their emotional involvement and problem-solving desires, and to just listen. Overall, this puts high demands on the closest family when adolescents have mental health problems. Given the importance of support from significant others [41], parent support programs provided by health and social services could be a good societal investment. However, it also needs to be acknowledged that adolescents who lack a supportive and safe home environment become particularly vulnerable. For these young people, other adults in their surroundings, such as teachers and sports coaches, could take on a more important role as support providers.

Close friends seem to be the most secure option for adolescent girls, and sometimes for boys, when considering help-seeking for mental health problems. While feeling understood and at ease may be helpful in the short term, the increased prevalence of mental health problems testifies to a situation where additional support is needed. The results show that adolescents experience difficulty in recognizing when their mental distress requires professional support. Promoting mental health literacy and help-seeking competency in the adolescent population in general is crucial to reduce self-blame and to help them discern when seeking help is pertinent. In addition, we propose new models of care that combine low-threshold support with high accessibility to meet the needs of adolescents as they transition into adulthood.

## Methodological considerations

The trustworthiness of a qualitative study relies on transparency and reflections about how sampling, data collection, and analysis were designed to address the research questions [60]. In this study, we wanted to explore adolescents' experiences and perceptions of help-seeking for mental health problems. Given that we wanted to capture adolescents' perceptions, attitudes, and norms in relation to help-seeking behaviour, FGDs were considered a suitable data collection method. Although individual interviews might yield richer insights into personal help-seeking experiences, they are unlikely to sufficiently capture the wider range of collective perceptions surrounding the topic [61]. The adoption of a GT design and, specifically, a SA can be considered a strength in this study. The iterative design – with two rounds of FDGs and interim analysis – strengthens the credibility and confirmability of the findings, by the triangulation of first-round analysis. The involvement of three researchers in the analysis further permitted a triangulation of investigators and strengthened the study's trustworthiness. The SA design was purposefully chosen because it goes beyond Classical Grounded Theory [62] to allow for an exploration of relationalities, positions, and discourses, rather than focusing solely on basic social processes. The production of the social world and situational maps [37] also align with Rabiee's [45] description of data interpretation for FDGs, where the researcher takes a step back to view the "big picture". We consider that this added depth to our analysis, along with an augmented dynamic to the exploration of a complex relational and social phenomenon. The application of GT involves going beyond the descriptive level to construct analytical categories that can be useful beyond the specific research situation. Hence, we believe that our constructed categories on how, respectively, close and distant emotional connectedness connect to help-seeking among adolescents may be transferable to other settings. However, this is for further research to explore and judge.

Moderator experience and skill are crucial to the performance of FDGs [61]. The first and last authors, who moderated the sessions, have extensive experience of qualitative research with adolescents and young adults, as well as within the field of mental health and illness. We chose the school setting to increase the sense of familiarity and to reduce the potential power imbalance between researcher and participant that might arise when the participant is invited to an unfamiliar setting. We started each session with an activity, such as choosing talking cards (first round) or completing a positional map (second round). This strategy ensures that every participant is engaged and heard at the start of the session, while reducing the risk of conformity [61]. Altogether, our impression of the FGDs was that all participants shared their perceptions and experiences in a safe and open discussion climate. The sampling strategy produced a sample of adolescents equally distributed between boys and girls and with diverse socioeconomic and cultural backgrounds. Given the increase in school absence among adolescents in Sweden over recent years, it should be noted that recruitment through schools omits absent pupils. As school absence is associated with elevated levels of mental health problems, this may have introduced a sampling bias and may be considered a weakness of the study. More broadly, the sampling approach may also be subject to common biases in qualitative research with adolescents, including self-selection bias, whereby individuals who feel more comfortable discussing mental health or who have a particular interest in the topic are more likely to participate, as well as the potential underrepresentation of more marginalized or hard-to-reach groups. Still, experiences of mental health problems or prior healthcare contact were not inclusion criteria, which may have contributed to capturing a broader range of perspectives.

## Conclusions

This study revealed a nuanced image of emotional connectedness in adolescents' help-seeking for mental health problems. The results add depth to the understanding of emotional connectedness as it relates to help-seeking in adolescence by providing rich descriptions of adolescents' perceptions of both close and emotionally distant relationships in the context of mental health problems. A dual image emerges at both ends of the spectrum, where both emotional closeness and distance can either facilitate or hinder help-seeking. Trust and comfort in close relationships generally offer a secure and predictable option, but can also be perceived as conditional and burdensome. At the same time, emotional distance plays

a more context-dependent role in adolescents' help-seeking. In some cases, relationships characterized by emotional distance, such as formal relationships with professionals, can facilitate help-seeking when underpinned by expectations of confidentiality and safety, particularly when close relationships are perceived as unavailable or unsuitable. Similarly, more informal distant relationships, such as with online contacts or parents of friends, may enable self-disclosure by offering a perceived space free from immediate personal consequences. However, the facilitating nature of emotional distance appears sensitive to general perceptions shared in networks and may act hindering when such relationships are experienced as unpredictable or when the potential outcomes of disclosure remain uncertain.

## Acknowledgments

We would like to thank the adolescents who participated in this study for their generosity, wisdom, and openness.

## Author contributions

**Conceptualization:** Katrin Häggström Westberg, Malin Eriksson, Mikael G Ahlborg.

**Data curation:** Katrin Häggström Westberg, Mikael G Ahlborg.

**Formal analysis:** Katrin Häggström Westberg, Malin Eriksson, Mikael G Ahlborg.

**Funding acquisition:** Katrin Häggström Westberg, Mikael G Ahlborg.

**Investigation:** Katrin Häggström Westberg, Mikael G Ahlborg.

**Methodology:** Katrin Häggström Westberg, Malin Eriksson, Mikael G Ahlborg.

**Project administration:** Mikael G Ahlborg.

**Resources:** Mikael G Ahlborg.

**Software:** Mikael G Ahlborg.

**Visualization:** Mikael G Ahlborg.

**Writing – original draft:** Katrin Häggström Westberg, Malin Eriksson, Mikael G Ahlborg.

**Writing – review & editing:** Katrin Häggström Westberg, Malin Eriksson, Mikael G Ahlborg.

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
