## [Decision Letter · Decision Letter 0]

5 Mar 2026

PMEN-D-25-00510

Emotional connectedness and adolescent help-seeking behavior for mental health problems: A situational analysis

PLOS Mental Health

Dear Dr. Ahlborg,

Thank you for submitting your manuscript to PLOS Mental Health and I am very sorry for the severe delay in reaching a decision. This was due to difficulties securing reviewers. Thank you for your patience. After careful consideration of the reviewer reports, we feel that your paper has merit but does not fully meet PLOS Mental Health’s publication criteria as it currently stands. Therefore, we invite you to submit a revised version of the manuscript that addresses the points raised during the review process.

Please address all comments, which you can find at the end of this email.

We look forward to receiving your revised manuscript.

Kind regards,

Karli Montague-Cardoso

Staff Editor

PLOS Mental Health

**Journal Requirements:**

i. Please clarify all sources of financial support for your study. List the grants, grant numbers, and organizations that funded your study, including funding received from your institution. Please note that suppliers of material support, including research materials, should be recognized in the Acknowledgements section rather than in the Financial Disclosure.

ii. State the initials, alongside each funding source, of each author to receive each grant. For example: "This work was supported by the National Institutes of Health (####### to AM; ###### to CJ) and the National Science Foundation (###### to AM)."

iii. State what role the funders took in the study. If the funders had no role in your study, please state: “The funders had no role in study design, data collection and analysis, decision to publish, or preparation of the manuscript.”

iv. If any authors received a salary from any of your funders, please state which authors and which funders.

2. Please ensure that your Ethics Statement is available in its entirety at the beginning of your Methods section, under a subheading 'Ethics Statement'.

3. Please provide separate figure files in .tif or .eps format.

**Additional Editor Comments (if provided):**

The evidence supports the therapeutic and communication tools, particularly Emotional connectedness, as potential facilitators of psychological well-being. These findings are grounded in the systematically identified study and provide a promising evidence base. Beyond these evidence-based insights, I Suggest the inclusion in the discussion of the following references:

Vale Lucas, C., & Soares, L. (2014). Being a teen and learning how to surf anxiety: Integrating narrative methods with cognitive–behavioral therapy. Journal of Poetry Therapy, 27(2), 69–82. https://doi.org/10.1080/08893675.2014.895489

Soares, L., Botella, L., Corbella, S., de Lemos, M. S. & Fernandéz, M., (Apr 2013), Diferentes estilos de clientes and construcción de la alianza con un terapeuta, Revista Argentina de Clinica Psicologica. 22, 1, p. 27-36 10 p.

Reviewers' comments:

Reviewer's Responses to Questions

**Comments to the Author**

1. Does this manuscript meet PLOS Mental Health’s publication criteria? Is the manuscript technically sound, and do the data support the conclusions? The manuscript must describe methodologically and ethically rigorous research with conclusions that are appropriately drawn based on the data presented.

Reviewer #1: Yes

Reviewer #2: Yes

2. Has the statistical analysis been performed appropriately and rigorously?

Reviewer #1: N/A

Reviewer #2: N/A

3. Have the authors made all data underlying the findings in their manuscript fully available (please refer to the Data Availability Statement at the start of the manuscript PDF file)?

Reviewer #1: No

Reviewer #2: Yes

4. Is the manuscript presented in an intelligible fashion and written in standard English?

Reviewer #1: Yes

Reviewer #2: Yes

5. Review Comments to the Author

Reviewer #1: Dear authors,

The study is timely and can contribute meaningfully to the existing literature. However, several issues need to be addressed.

The current study focuses on teenagers, a population that differs substantially from adults in terms of developmental stage, cognitive maturation, emotional regulation, social capital formation, and the capacity to mobilize such capital. While the manuscript references social connections and loneliness among young people, these discussions remain relatively surface-level and insufficiently anchored in developmental theory.

The authors should more explicitly situate the study within the psychological and sociological distinctions between adolescence and adulthood. Adolescence is characterized by identity formation, heightened peer orientation, ongoing neural development (particularly in socio-emotional and executive systems), and evolving autonomy from family structures. These features greatly shape how social connectedness is experienced, interpreted, and enacted.

Clarifying these distinctions would significantly strengthen the manuscript’s contribution by demonstrating why findings derived from teenagers cannot simply be extrapolated from adult populations. This clarification should be reflected not only in the literature review but also in the framing of the study’s objectives, the title, and the abstract to foreground the theoretical significance of the work.

Moreover, adolescence is a critical period for developing durable patterns of social connectedness. Therefore, the manuscript would benefit from a deeper discussion of how the Swedish socio-cultural context may shape this process. Without such contextualization, the findings risk appearing decontextualized or culturally neutral. Integrating socio-cultural discussion would not only strengthen the study’s validity but also help readers understand how situational analysis helps embed the findings into broader structural conditions. This would also help expand the discussion section by connecting micro-level perceptions to macro-level institutional and cultural influences.

The methodology section requires a more detailed description of the participants and study sites. The description should explicitly justify how the sample reflects the broader adolescent population in Sweden. If representativeness cannot be ensured, this limitation should be acknowledged clearly.

Furthermore, methodological limitations need to be discussed more explicitly. Potential constraints may include sampling bias, self-report bias, cultural specificity, limited generalizability, or analytical constraints. Addressing these issues transparently will enhance the study’s integrity and credibility.

Finally, the uncertainty–absurdity nexus is omnipresent in adolescents’ lives, with important implications for mental health, the development and erosion of social capital, and patterns of help-seeking. The arguments and insights surrounding this nexus are articulated in a systematic, analytically rigorous manner that preserves both reliability and conceptual generality [1]. Engaging with this framework would help clarify adolescents’ perceptions of the relationship between help-seeking for mental health concerns and emotional connectedness, thereby strengthening the study’s theoretical coherence and enriching the discussion.

[1] Nguyen, M. H. (2026). Uncertainty versus Absurdity: Exploring the Propositions of Wild Wise Weird. https://books.google.com/books?id=Y-3AEQAAQBAJ

I hope these suggestions are helpful.

Cheers,

Reviewer #2: The manuscript addresses an important and timely issue. Its focus on relational dynamics offers a potentially valuable contribution to adolescent mental health research, particularly given the longstanding emphasis on individual level barriers rather than relational meaning making processes. However, there are conceptual, methodological, and analytic areas that require further clarification and strengthening before the manuscript would meet the standards of theoretical and methodological rigour.

Conceptual Framing. Emotional Connectedness vs. Perceived Meaning

The manuscript positions “emotional connectedness” as the key construct underpinning help-seeking behaviour. However, the findings suggest that what is most important is not connectedness per se, but adolescents’ interpretation of relational dynamics particularly around trust, confidentiality, predictability, and perceived burden. From a critical realist perspective, it is important to distinguish between the relational structure (closeness/distance), and the adolescent’s interpretation of that structure. Emotional closeness is at times treated as a relational property and at other times as a subjective experience imbued with meaning. Greater conceptual clarity would strengthen the manuscript. Are you analysing relational configurations, or adolescents’ causal beliefs about what makes help-seeking safe or unsafe? Relatedly, the manuscript would benefit from stronger engagement with existing help-seeking theory. Without this integration, the contribution risks appearing descriptive rather than theoretically generative.

Sample and Transferability

The study includes 24 adolescents aged 15 from two schools in one Swedish region. While appropriate for qualitative exploration, the manuscript would benefit from clarification regarding mental health status (self-identified distress? clinically assessed?) and discussion of how cultural context may shape relational norms and help-seeking.

Closeness and Burden

One of the most interesting findings is that emotional closeness can inhibit help-seeking due to fears of burdening others or threatening valued relationships. This resonates strongly with research demonstrating that adolescents’ help-seeking is shaped by perceived consequences for self and others, not merely availability of support. However, this point is not fully theorised in the discussion. Currently, the manuscript stops short of articulating the broader implications of this relational ambivalence.

Emotional Distance as Facilitative.

The finding that emotional distance can facilitate help-seeking is interesting. However, the manuscript should clarify if adolescents are distinguishing between emotional intimacy and structural safety. There is a risk that “distance” becomes an umbrella term encompassing multiple mechanisms like anonymity, reduced relational risk, and formal roles. These mechanisms should be disentangled.

Help-Seeking Intention vs Behaviour

The manuscript refers repeatedly to help-seeking intention. It would be important to clarify whether participants were describing hypothetical scenarios or reflecting on actual past behaviour. This distinction matters, particularly given well established intention behaviour gaps in help-seeking research.

Contribution to Knowledge

The manuscript’s key contribution appears to be the argument that neither emotional closeness nor distance is inherently facilitative of help-seeking, rather, the relational meaning attached to each determines its influence. This is a nuanced and valuable insight. However, the discussion would benefit from sharper articulation of what is genuinely novel here and how does this extend existing literature? It would also be useful to reflect on potential implications for intervention design. For example, interventions that simply encourage “talking to someone close” may be misguided if closeness is perceived as conditional or burdensome. This practical implication deserves stronger emphasis.

6. PLOS authors have the option to publish the peer review history of their article (what does this mean?). If published, this will include your full peer review and any attached files.

**Do you want your identity to be public for this peer review?** For information about this choice, including consent withdrawal, please see our Privacy Policy.

Reviewer #1: No

Reviewer #2: **Yes:**Dr Alisha O'Neill

Figure Resubmissions:

---

## [Decision Letter · Decision Letter 1]

1 May 2026

Emotional connectedness and adolescent help-seeking behavior for mental health problems: A situational analysis

PMEN-D-25-00510R1

Dear Mr. Ahlborg,

We are pleased to inform you that your manuscript 'Emotional connectedness and adolescent help-seeking behavior for mental health problems: A situational analysis' has been provisionally accepted for publication in PLOS Mental Health.

Best regards,

Karli Montague-Cardoso

Staff Editor

PLOS Mental Health

Reviewer Comments (if any, and for reference):

Reviewer's Responses to Questions

**Comments to the Author**

1. If the authors have adequately addressed your comments raised in a previous round of review and you feel that this manuscript is now acceptable for publication, you may indicate that here to bypass the “Comments to the Author” section, enter your conflict of interest statement in the “Confidential to Editor” section, and submit your "Accept" recommendation.

Reviewer #1: All comments have been addressed

2. Does this manuscript meet PLOS Mental Health’s publication criteria? Is the manuscript technically sound, and do the data support the conclusions? The manuscript must describe methodologically and ethically rigorous research with conclusions that are appropriately drawn based on the data presented.

Reviewer #1: Yes

3. Has the statistical analysis been performed appropriately and rigorously?

Reviewer #1: Yes

4. Have the authors made all data underlying the findings in their manuscript fully available (please refer to the Data Availability Statement at the start of the manuscript PDF file)?

Reviewer #1: No

5. Is the manuscript presented in an intelligible fashion and written in standard English?

Reviewer #1: Yes

6. Review Comments to the Author

Reviewer #1: All my comments have been addressed.

7. PLOS authors have the option to publish the peer review history of their article (what does this mean?). If published, this will include your full peer review and any attached files.

**Do you want your identity to be public for this peer review?** For information about this choice, including consent withdrawal, please see our Privacy Policy.

Reviewer #1: No
